# Differential Leukocyte and Platelet Profiles in Distinct Models of Traumatic Brain Injury

**DOI:** 10.3390/cells10030500

**Published:** 2021-02-26

**Authors:** William Brad Hubbard, Meenakshi Banerjee, Hemendra Vekaria, Kanakanagavalli Shravani Prakhya, Smita Joshi, Qing Jun Wang, Kathryn E. Saatman, Sidney W. Whiteheart, Patrick G. Sullivan

**Affiliations:** 1Spinal Cord and Brain Injury Research Center (SCoBIRC), University of Kentucky, Lexington, KY 40536, USA; bradhubbard@uky.edu (W.B.H.); hemendravekaria@uky.edu (H.V.); k.saatman@uky.edu (K.E.S.); 2Department of Physiology, University of Kentucky, Lexington, KY 40508, USA; 3Department of Neuroscience, University of Kentucky, Lexington, KY 40508, USA; 4Lexington Veterans’ Affairs Healthcare System, Lexington, KY 40502, USA; whitehe@uky.edu; 5Department of Molecular and Cellular Biochemistry, University of Kentucky, Lexington, KY 40536, USA; meenakshi.banerjee@utah.edu (M.B.); shravani.prakhya@uky.edu (K.S.P.); smita.joshi@uky.edu (S.J.); 6Department of Ophthalmology and Visual Sciences, University of Kentucky, Lexington, KY 40536, USA; qingjun.wang@uky.edu

**Keywords:** neutrophil, platelet-neutrophil aggregate, bioenergetics, Seahorse, respirometry, controlled cortical impact, mild TBI

## Abstract

Traumatic brain injury (TBI) affects over 3 million individuals every year in the U.S. There is growing appreciation that TBI can produce systemic modifications, which are in part propagated through blood–brain barrier (BBB) dysfunction and blood–brain cell interactions. As such, platelets and leukocytes contribute to mechanisms of thromboinflammation after TBI. While these mechanisms have been investigated in experimental models of contusion brain injury, less is known regarding acute alterations following mild closed head injury. To investigate the role of platelet dynamics and bioenergetics after TBI, we employed two distinct, well-established models of TBI in mice: the controlled cortical impact (CCI) model of contusion brain injury and the closed head injury (CHI) model of mild diffuse brain injury. Hematology parameters, platelet-neutrophil aggregation, and platelet respirometry were assessed acutely after injury. CCI resulted in an early drop in blood leukocyte counts, while CHI increased blood leukocyte counts early after injury. Platelet-neutrophil aggregation was altered acutely after CCI compared to sham. Furthermore, platelet bioenergetic coupling efficiency was transiently reduced at 6 h and increased at 24 h post-CCI. After CHI, oxidative phosphorylation in intact platelets was reduced at 6 h and increased at 24 h compared to sham. Taken together, these data demonstrate that brain trauma initiates alterations in platelet-leukocyte dynamics and platelet metabolism, which may be time- and injury-dependent, providing evidence that platelets carry a peripheral signature of brain injury. The unique trend of platelet bioenergetics after two distinct types of TBI suggests the potential for utilization in prognosis.

## 1. Introduction

Traumatic brain injury (TBI) continues to be a significant world-wide clinical problem that affects millions of individuals each year. While there remains a lack of FDA-approved pharmaceuticals for TBI treatment, the on-going understanding of the underlying pathobiology of head trauma not only identifies therapeutic targets but also pinpoints new avenues for TBI prognosis. Distinct pathological hallmarks, such as mitochondrial deficits and vascular damage, have been identified as major mechanisms of TBI [1,2,3,4,5,6,7]. Early blood–brain barrier (BBB) impairment propagates interaction and infiltration of blood cells within the brain parenchyma, a process that is tightly regulated by the highly selective barrier under normal conditions [8]. After TBI, platelets and leukocytes attach to the vascular walls and aggregate in the injured cerebral tissue while also inducing inflammatory and immune responses [9,10,11]. Indeed, we have described BBB dysfunction following TBI as the epicenter for mechanisms of inflammation and thrombosis, which are mediated in part by platelets and leukocytes [12]. Thromboinflammation is among a variety of systemic pathologies that are now widely appreciated as potential consequences of TBI.

Alterations of platelet physiology have been reported clinically following severe TBI. Platelet dysfunction, characterized by inhibition of platelet arachidonic acid (AA) and ADP receptors leading to lower levels of AA/ADP-induced platelet activation, occurs early after severe TBI without major systemic trauma [13,14]. This inhibition appears to be dependent on TBI severity, with mild TBI demonstrating lower levels of platelet receptor inhibition [15]. TBI, by itself, results in increased bleeding time, or hypocoagulation, early after injury, which is indicative of deficits in coagulation pathways or platelets’ ability to activate [14]. TBI also results in AA platelet receptor downregulation in the first few days following injury [16]. Importantly, there are shifts between hypocoagulable and hypercoagulable states following TBI [17]. Preclinical studies recapitulated these findings in murine models of moderate-to-severe TBI. These studies demonstrate immediate hypocoagulation followed by delayed hypercoagulation events [18,19,20,21,22]. Recently, one preclinical study showed changes acutely after closed head injury (CHI), demonstrating hypercoagulability and increased aggregation of platelets at 6 h after CHI [19]. It is important to understand platelet dynamics to fully appreciate mechanisms of thromboinflammation.

Several neurodegenerative diseases and pathologies have utilized platelet function as a biomarker of injury progression. Platelet respiration is an emerging pathological biomarker as it can serve as a systemic marker for underlying mitochondrial dysfunction in metabolic tissues, such as the brain [23]. The concept that mitochondrial function in platelets can exemplify deficits in cerebral bioenergetics during neurological disease progression is not entirely new [24]; early work showed that Complex I activity is reduced in platelets of patients with idiopathic Parkinson’s disease [25]. Furthermore, this has been appreciated in the context of Alzheimer’s disease (AD) through decreased Complex IV activity in platelets of AD patients [26,27]. Maximal respiratory capacity in platelets correlates with respiratory control ratio (RCR) of mitochondria in skeletal and cardiac muscle [28] in addition to the brain [29]. Furthermore, changes in platelet coupling efficiency after cardiac arrest have been shown to represent underlying cerebral bioenergetic dysfunction [30]. We have shown acute mitochondrial impairment following both single and repeated mild TBI [2] and metabolic deficiencies that continue long-term following mild TBI [1]. Although platelet respiration shows promise as a peripheral monitor of neurodegenerative disease, no research, to date, has examined platelet bioenergetics within the context of TBI.

TBI presents clinically in a highly heterogeneous fashion, in terms of severity, symptomatic onset/progression, and underlying pathology. Even TBIs that are classified under the same Glasgow Coma Scale (GCS) score vary widely in terms of underlying pathology [31]. The pathologies and secondary injury mechanisms consist of multiple endophenotypes, such as bioenergetic dysfunction, vascular damage, and thromboinflammation. Therefore, injury-associated markers are necessary to systematically monitor these endophenotypes and, therefore, specific aspects of neurologic injury and recovery. This is also crucial in the context of repetitive mild head injury. Since repeated mild TBI results in worsened or prolonged outcomes [32], it is pertinent to track recovery following a mild TBI. Current return-to-play and return-to-duty guidelines for athletes and military service members, respectively, are put in place to prevent the additive effects of head trauma [33]. These are based upon memory recall and concentration compared with baseline assessments. However, cognitive recovery and symptom resolution do not necessarily equate to neuronal recovery [34]. Therefore, objective biomarkers that mirror deficits in cerebral metabolism after mild TBI are needed to assess sufficient brain recovery. We, therefore, focused on platelet function to further our understanding of metabolic alterations systemically after head trauma.

There are several characterized preclinical models of TBI that focus on replicating certain distinct aspects of clinical TBI. In the current study, we focus on head injury models that induce brain injury via compressive mechanical force. Other preclinical models, such as the closed-head impact model of engineered rotational acceleration (CHIMERA), involve shear deformation forces generated by rapid acceleration/deceleration of the head, which is a component of human TBI. Nevertheless, these compressive injury models indeed replicate clinical aspects of TBI-induced outcomes, such as behavioral deficits, axonal injury, inflammation, and metabolic dysfunction [2,35,36,37,38]. The controlled cortical impact (CCI) model involves impact directly to the dura of the brain producing a relatively defined, focal cortical contusion that results in cortical and hippocampal neurodegeneration at the site of injury. CCI can be graded in terms of severity by modifying the parameters of impact, namely depth of piston entry into the brain parenchyma. CCI produces mitochondrial dysfunction, neuroinflammation, neuronal death, and intracerebral hemorrhage. The CHI model involves a midline impact to the skull, which generates a relatively diffuse brain injury that produces neuroinflammation and deficits in brain metabolism. CHI typically does not produce overt brain hemorrhage or cell death.

To examine blood cell changes after TBI, we utilize these two distinct models of experimental brain injury to monitor the acute profile of leukocyte and platelet activity. We hypothesize that blood cell dynamics are altered transiently following TBI, in a manner dependent upon injury model. Furthermore, alterations in blood cell dynamics can impart understanding of mechanisms of thromboinflammation and coagulopathy.

## 2. Materials and Methods

### 2.1. Experimental Design

All of the studies performed were approved by the University of Kentucky Institutional Animal Care and Use Committee (IACUC). Additionally, the Division of Laboratory Animal Resources at the University is accredited by the Association for the Assessment and Accreditation for Laboratory Animal Care, International (AAALAC, International) and all experiments were performed within its guidelines. All animal experiments complied with ARRIVE (Animal Research: Reporting of In Vivo Experiments) guidelines and experiments were carried out in accordance with the National Institutes of Health Guide for the Care and Use of Laboratory Animals (NIH Publications No. 8023, 8th edition, revised 2011). All experiments were conducted using male C57BL/6J mice (2–3 months old; Jackson Laboratories, Bar Harbor, ME).

Animals were randomly assigned to groups and all data analyses were performed blinded to treatment groups. Experimental groups were euthanized at either 6 or 24 h after TBI or sham injury. The animals were housed five per cage and maintained in a 14 h light/10 h dark cycle. All animals were fed a balanced diet ad libitum and water was reverse osmosis-generated. For all outcomes, experiments were conducted with biological replicates of n = 5–10/group. Additionally, technical triplets were used in each assay.

### 2.2. Controlled Cortical Impact

The CCI procedure was performed according to past studies [35,36]. Prior to injury, animals were anesthetized using 2–5% isoflurane and their heads were shaved and placed in a Kopf stereotaxic frame for proper positioning under a pneumatic impactor (TBI-0310 Impactor, Precision Systems and Instrumentation (PSI), Fairfax, VA, USA). The animals’ body temperature was maintained at 37 °C with the use of an isothermal pad. Artificial tears were placed in their eyes and anesthesia was maintained at 2–3% isoflurane during the procedure. Prior to scalp incision, the scalp was cleaned with betadine and 200 µL of a 10% local anesthetic in saline solution (Sensorcaine-MPF (Bupivacaine HCl) 0.5% (Henry Schein Animal Health, Dublin, OH, USA) and Epinephrine 1:200,000) was injected subcutaneously at the site. A 3-mm craniotomy was performed lateral to the midline on the left side of the skull centered between lambda and bregma, without disrupting the dura. CCI groups received a unilateral injury, classified as severe (1.0 mm depth of contusion at 3.5 m/sec with a dwell time of 500 msec), directly to the surface of the brain using a 2 mm impactor tip. Sham animals received a craniotomy but did not receive an impact to the brain. Following the injury, the craniotomy was covered with an absorbable hemostat (Surgicel), followed by a sterilized plastic surgical cap, and incisions were then closed with surgical staples. Mice were taken off isoflurane exposure and were removed from the stereotaxic frame. Mice were then placed in a clean cage that was temperature-controlled at 37 °C with a heating pad until the animals were mobile and fully responsive.

### 2.3. Closed Head Injury

Mice were subjected to CHI based on previous studies [2,39]. Following induction of anesthesia in a chamber using 3% isoflurane for 1–2 min and shaving of the head, mice were transferred into a stereotaxic frame with non-rupture Zygoma ear cups (Kopf, Instruments, Tujunga, CA, USA) where anesthesia was maintained using 2.5% isoflurane delivered via a nose cone. The surgical area of the scalp was cleaned with a betadine solution. Following injection of 0.2 mL 1:200,000 epinephrine and 0.5% bupivacaine in sterile, normal saline for local analgesia, the scalp was resected. Mice then received a mild CHI using a pneumatically controlled cortical impact device (TBI-0310 Impactor, PSI) with a custom made, 5 mm diameter, pliant, silicone tip with a hardness of 55 Shore A. The tip was aligned along the midline suture between the bregma and lambda sutures with the edge of the tip lined up with the lambda suture. The device was programmed to impact at an intended depth (2.0 mm) at a 3.5 m/sec velocity with a 500 msec dwell time. Sham-injured mice received anesthesia and underwent the same surgical procedure without receiving an impact. The total duration of anesthesia was controlled for all animals to be 10 min. To assess loss of consciousness, mice were immediately removed from the stereotaxic device after impact and placed onto a heating pad on their backs. Apnea duration and time to right spontaneously to a prone position (righting reflex) were assessed. After righting, mice were briefly re-anesthetized to suture their scalps and returned to the heating pad to fully recover. Mice were injected (subcutaneous; s.c.) with 1 mL of sterile, normal saline to increase hydration, encouraging a normal eating response post-injury and maintenance of healthy weight.

### 2.4. Mouse Platelet Isolation

At either 6 h or 24 h post-injury, animals were asphyxiated with CO_2_, and up to 1 mL blood was collected via cardiac puncture. Blood was harvested using final concentrations of 0.38% sodium citrate solution (supplemented with 0.2 U/mL apyrase and 10 ng/mL prostacyclin (PGI_2_) (Cayman Chemical, Ann Arbor, MI, USA; Cat #18220)) in a 26G 3/8 syringe. Approximately 100 µL of whole blood was taken for hematology counts and platelet-neutrophil aggregation (PNA) assays, while the remaining volume was used for isolating platelets for Seahorse analysis. The 2 mL tube, containing blood for Seahorse analysis, was centrifuged at 100× *g* for 10 min at room temperature (RT; 20 to 25 °C). After centrifugation, the top yellow layer (platelet-rich plasma; PRP) was removed, being careful not to take the thin buffy coat layer or dark red RBC layer, and transferred into 1.5 mL tubes. The 1.5 mL tubes were centrifuged at 100× *g* for 10 min at RT. The supernatant (PRP) was removed and transferred to 1.5 mL tubes. Prostacyclin (PGI_2_; 30 µL) was freshly added from a stock solution (prepared as 1 mg/mL PGI_2_ in 50 mM Tris, pH 9.5) to a phosphate/glucose buffer (50 mL 1X PBS, 45 mg glucose (5 mM)) at time of assay. This phosphate-prostaglandin-glucose buffer (PPG) was added to increase the volume of PRP four-fold and mixed gently with a pipette. The samples were centrifuged at 2000× *g* for 1 min at RT. The tubes were then gently rotated and centrifuged again 2000× *g* for 5 min at RT. The supernatant (plasma) from this spin was removed with a pipette. The remaining platelet pellets were resuspended in 1 mL of PPG buffer and then the tubes were topped off with PPG buffer. The tubes were then centrifuged at 2000× *g* for 1 min at RT, rotated and centrifuged again at 2000× *g* for 2 min at RT. The supernatant from this last spin was removed and discarded and the pellet resuspended in 50 µL PPG buffer. Protein concentration was determined using BCA protein assay kit (Pierce, Cat # 23,227) recording the absorbance at 560 nm on Biotek Synergy HT plate reader (Winooski, VT, USA). Approximately 1 µg platelet protein corresponds to ~1 × 10^6^ platelets, for which platelet counts were measured using a Z2 Coulter Counter (Beckman Coulter, Inc., Miami, FL, USA).

### 2.5. Hematology Analysis

Of the aliquoted 100 µL of whole blood, 50 µL was used to perform whole blood counts, including red blood cells, white blood cells (leukocytes), platelets, and mean platelet volume, using the IDEXX ProCyte Dx analyzer (IDEXX Laboratories, Westbrook, ME, USA).

### 2.6. Flow cytometry—Platelet-Neutrophil Aggregation

The remaining 50 µL of whole blood was taken for fluorescence-activated cell sorting (FACS) studies. This assay was performed on live cells the day of blood collection. Whole mouse blood was incubated with FITC-anti-CD41/61 (platelets) and APC-anti-Ly6G (neutrophils) antibodies for 30 min at RT (see Appendix A for gating strategy). To lyse the red blood cells, 1 mL of 1X BD FACS/Lyse Solution (BD Biosciences, San Jose, CA, USA; Cat No. 349202) was added to samples and incubated for 15 min at RT in dark. From this, 100 µL was resuspended in 1 mL of PBS pH 7.4 and samples were analyzed on the BD LSRIITM flow cytometer (BD Biosciences) and 500,000 events were acquired per sample with gating around 50,000 granulocytes. Data were then analyzed using the FlowJo software (v7.6.5; BD Biosciences). Quantification shows PNAs as measured by double-positive events for both CD41/61 and Ly6G. Platelet-neutrophil dynamics were measured as: (% double-positive events at 24 h post-injury) minus (% double-positive events at 6 h post-injury).

### 2.7. Measurement of Mouse Platelet Bioenergetics

One previous report demonstrated that pooling whole blood from up to five mice would provide sufficient platelet numbers to examine respiration [40]. To our knowledge, this is the first report demonstrating that platelet respiration from a single mouse can be measured using the Seahorse XFe96 technology. Adequate resolution at low oxygen consumption rates (OCRs) allows as little as 10 µg platelet protein (approximately 1 × 10^7^ platelets) to be used in each well for oxygen consumption measurements [41].

Measurements of bioenergetics in platelets were completed using a Seahorse XFe96 Flux Analyzer (Agilent Technologies, Santa Clara, CA, USA). The OCR was measured in the presence of substrates, inhibitors, and uncouplers related to OXPHOS as modified from previously described methods [42,43]. The day before the planned experiment, the sensor cartridge of Extracellular Flux kit was filled with 200 µL deionized water and kept at 37 °C overnight. At least 30 min prior to use, the water was removed, the plate filled with 200 µL XF calibrant solution and put back at 37 °C. The injection ports, A to D of the sensor cartridge, were loaded separately or in combinations of substrates/inhibitors/uncouplers to measure different states of respiration. Before loading, the stocks were diluted appropriately in PPG buffer such that after each sequential injection, the final concentration of the modulators was 5 mM pyruvate, 2.5 mM malate, and 10 mM succinate (via Port A), 5 µM oligomycin A (via Port B), 4 µM FCCP (via Port C), and 1 µM antimycin A (via Port D). Either 10 or 30 µg total platelet protein (approximately 1 × 10^7^ or 3 × 10^7^ platelets) was loaded per well on a Seahorse XFe96 assay plate in a total volume of 30 µl (platelet sample plus PPG buffer). The assay plates were centrifuged at 2100× g for 4 min at RT. Additional pre-warmed PPG buffer was added to bring the starting volume to 175 µL. After the calibration step, the utility plate was replaced by the assay plate carrying platelet samples. Seahorse Standard XFe96 flux assay plates were utilized for platelet analysis. Oxygen consumption rates (OCR) and extra-cellular acidification rate (ECAR) were recorded in the absence or the presence of various substrates/inhibitors which were added from port A to port D. These measurements were performed on intact (non-permeabilized) platelets. Basal readings were performed in the presence of glucose in the buffer. OXPHOS, LEAK, MAX, and non-mitochondrial (NON-MITO) oxygen consumption respiration rates were recorded based on subsequent port injections (Figure 1). Basal glycolysis was measured as ECAR in the presence of glucose without substrate addition. Maximum glycolysis was measured as ECAR after the addition of oligomycin according to the injection paradigm in Figure 1. Platelet coupling efficiency was calculated by OXPHOS/LEAK, while glycolytic reserve capacity was calculated by Max glycolysis-Basal glycolysis. Raw OCR and ECAR values were used for analysis within a given experiment and reported in all figures.

### 2.8. Statistical Analysis

Power analysis was conducted for experimental data a priori based on an effect size and expected data variance. Statistical analysis was performed using Graph Pad Prism 8 (GraphPad Software, San Diego, CA, USA) or JMP Pro 14 (SAS, Cary, NC, USA). For all analyses, the significance of differences among groups was set at *p* < 0.05. For each measure, data were measured using interval/ratio scales. The Brown–Forsythe and Bartlett’s tests were performed to ensure homogeneity of variance. Furthermore, the Shapiro–Wilk test was completed to ensure normality. As these criteria were met for all experimental data, parametric statistics were employed for all analyses. For data obtained on PNAs, a statistical blocking factor for assay day was utilized to reduce within group variability between assays runs. For platelet bioenergetics data, raw OCR/ECAR values were used for analysis within a given experiment. For all analyses, unpaired two-tailed t-test was performed to compare sham and injured groups.

## 3. Results

### 3.1. CCI Does Not Change Platelet Parameters But Decreases Early Blood Leukocyte Levels

To assess the acute effects on peripheral blood cell profiles following CCI, we examined hematological parameters at 6 and 24 h post-injury. There were no significant differences, either at 6 or 24 h, for RBC counts (Figure 2). Leukocyte counts were significantly decreased at 6 h post-CCI compared to sham (Figure 2). Systemic leukocyte counts were restored by 24 h post-injury. There were also no changes in platelet counts or mean platelet volume (MPV) early after CCI.

### 3.2. CHI Does Not Alter Platelet Parameters But Rather Increases Blood Leukocyte Levels

Similar to our previous reports [37], a single CHI resulted in an immediate apnea response (17.2 ± 3.0 s) and an increase in time to right after a 10-minute sustained bout of anesthesia (CHI: 5.0 ± 1.3 min vs. Sham: 1.3 ± 0.1 min). RBC counts were unchanged early after CHI (Figure 2). However, we found significantly higher leukocytes counts in the blood following CHI at both 6 and 24 h post-injury, compared to sham. This may be an indication of a delayed immune response in the brain or leukocytosis induced by diffuse mild TBI. We show that platelet counts and MPV are not changed at either 6 or 24 h following CHI.

### 3.3. CCI Causes a Dynamic Shift in Early Platelet-Neutrophil Aggregation

Platelet-leukocyte aggregates are indicative of the thromboinflammatory response following cerebral trauma [44]. Platelet-neutrophil aggregates (PNAs) represent the vast majority of platelet-leukocyte aggregates; as such, PNAs have been shown to increase following acute ischemic stroke [45]. Although we find that PNA levels are not significantly different at either 6 or 24 h after CCI compared to sham, the change in PNAs between these early time points was significantly different between sham and CCI (Figure 3).

### 3.4. CHI Does Not Alter Platelet-Neutrophil Aggregation Acutely After Injury

Although at both 6 and 24 h the mean PNA levels were higher for the CHI group than the sham controls, this difference did not reach statistical significance (Figure 3). This slight increase may be indicative of an early thromboinflammatory response following CHI. There was no early shift in PNA levels between 6 and 24 h following CHI.

### 3.5. CCI Produces an Acute Shift in Platelet Coupling Efficiency

Metabolic changes in platelets, which may relate to disease progression [23], can be readily assessed using respirometry techniques. Platelet respiratory parameters, such as coupling efficiency and reserve capacity, were similar to previously published results [42]. As stated previously, this is the first study, to our knowledge, that demonstrates the measurement of platelet respiration from a single mouse (versus pooled samples). There were no changes in any platelet respiration state at either 6 or 24 h following CCI (Figure 4). However, the coupling efficiency of platelets was lower at 6 h and higher at 24 h after CCI compared to sham (Figure 4). An expanded time course showed that coupling efficiency returned to sham levels at 12 and 18 h post-injury before an increase at 24 h following CCI (Appendix A), demonstrating a gradual transition acutely after CCI. The glycolytic signature of platelets was unchanged after CCI (Figure 4).

### 3.6. CHI Causes Early Alterations in Platelet Bioenergetics after Injury

After CHI, platelets displayed lower OXPHOS levels at 6 h post-injury compared with sham (Figure 4). By 24 h, however, platelets from mice with CHI exhibit significantly higher MAX respiration levels compared to those from sham mice (Figure 4). Neither platelet coupling efficiency nor platelet glycolysis (Figure 4) changed acutely after CHI.

## 4. Discussion

This study shows that early blood cell dynamics are differentially altered between experimental contusion TBI and mild diffuse TBI. The two distinct injury models used in this study represent differing injury severity, pathobiology, and spatial injury progression. The immune response following CNS injury is initiated by damage- and pathogen-associated molecular patterns (DAMPs/PAMPs), followed closely by neutrophil interactions in the brain [46]. The exact progression of leukocyte mobilization or brain infiltration is not clear, but we show in this study that these mechanisms are distinct between closed head impact and contusion brain injury. Waves of early leukocyte infiltration into the brain, including neutrophils (hours after injury) and monocytes/T lymphocytes (days after injury), are well-documented following CCI [9,44,47]. Comparatively, leukocyte infiltration is modest following CHI [10]. Our results show an early, transient decrease in leukocyte counts after CCI. We also demonstrate that blood leukocyte levels increase at both 6 and 24 h after CHI compared to sham. Clinically, blood leukocyte levels are increased early after TBI [48] and this increase is dependent, to some extent, on injury severity [47]. Elevated blood leukocyte levels, measured upon hospital admission, were identified as a prognostic biomarker for severe head injury by correlating with an unfavorable outcome at 6 months post-injury [49], though another study found no association with increased leukocyte counts and 6 month outcome [50]. Blood leukocyte levels measured following CCI do not align with these clinical findings. It has been shown that catecholamine released after TBI contributes to leukocytosis, including mobilization of marginal or adherent leukocytes [51,52]. CHI may induce leukocytosis, potentially via catecholamine release without early overt infiltration into the brain. The large increases that we observed in blood leukocyte population after CHI could be the physiologic mechanism for on-going or even delayed immune response in the injured brain and are clinically relevant. Longitudinal studies to expand upon the current results are needed.

Platelet-leukocyte interactions increase in environments of inflammation. It is well-known that the influx of inflammatory blood cells occurs at the BBB after brain damage [6,12] and this infiltration, which is propagated by the dysfunctional neurovascular unit, contributes to on-going neuroinflammation [53,54]. In the current study, we demonstrate how TBI can contribute to early systemic inflammation as well. Clinically, acute platelet-leukocyte aggregation increases have been correlated with neurological deterioration following ischemia [45]. Post-TBI microthrombi have been observed in the brain; indeed, platelet-leukocyte interactions have been shown to promote thrombosis following TBI [55]. We show a small, non-significant increase in PNA levels at 6 h after CCI with a significant decline from 6 to 24 h post-CCI. These acute PNA dynamics need further investigation to understand their role in pathophysiology following CCI. Following CHI, slight, though non-significant, increases in PNAs were observed acutely, which could be related to elevated leukocyte levels in the blood after CHI. Future studies with extended time course, and incorporation of repeated CHIs, are warranted to examine the temporal progression of PNAs after TBI. Furthermore, platelets have been shown to propagate monocyte differentiation into macrophages [56], so platelet interaction with monocyte/macrophage may also be important following TBI and, in turn, influence the bioenergetic status of platelets.

While neuroimaging techniques, including computed tomography (CT), MRI, diffusion-weighted imaging, and diffusion tensor imaging, are commonly used for diagnosis and prognosis of TBI, these are not meant for point-of-care or longitudinal tracking. Serum and cerebrospinal fluid (CSF) markers also are routinely used for diagnosis and prognosis of TBI. As an alternative, platelets have been widely recognized to mirror the progression of neurological disease. As the pathobiology of TBI is dynamic, these biomarker-based measures are just a glimpse of a continually evolving disease state. While serum biomarkers may have complex kinetics (release from the brain, crossing the BBB, and systemic clearance), platelet function can be pathologically altered following interactions in the damaged brain. We present initial findings of platelet alteration after TBI, but additional research is needed. Indeed, future studies can be performed to monitor platelet function with FACS-based techniques (collected serially) correlated to behavioral and histological outcome as a “within subject” analysis. Additionally, RNA-seq could be employed to build off of this work enabling biomarker discovery in the blood transcriptome following TBI [57].

We show here that platelet counts and MPV are stable early after either CCI or CHI. Previous studies showed that platelet counts are unaltered at 6 h following weight-drop closed head impact [19]. Another study found no changes in platelet counts or MPV after closed head TBI in the rat within 1 h after injury [18]. Hypercoagulability occurs early following CHI, despite no alteration in platelet count [19]. Our results build upon these findings and further demonstrate that changes in platelet function or metabolism are independent of changes in platelet count or MPV.

To our knowledge, no study has been published on platelet bioenergetic changes after TBI. We show distinct temporal alterations in platelet respiration in a model of mild TBI at acute time points. It is known that platelet activation leads to increased oxidative phosphorylation [58]. For example, the addition of thrombin increases OXPHOS, while the addition of indomethacin inhibits platelet activity and decreases OXPHOS. After closed head injury, we show that platelet respiration increases at 24 h, alluding to elevated platelet activation and potentially mechanisms of delayed thrombosis after CHI. Following contusion brain injury, platelet coupling efficiency is decreased at 6 h and increased at 24 h after CCI.

Platelets are uniquely suited to play a variety of roles following TBI. Platelets, measuring ~1–3 µm, have the ability to circulate in a majority of the brain microvasculature, facilitating their critical role in vascular remodeling following injury [59,60]. As microvascular injury is a hallmark of TBI, there is high potential for enhanced platelet-brain cell communication in these compromised areas [5]. Platelets may play a major role in BBB dysfunction after TBI. In areas of BBB breakdown, the glycoprotein Ib-IX-V complex on the surface of platelets binds to von Willebrand factor (vWF) in the subendothelium [61]. The role of platelets goes well beyond that related to hemostasis following TBI; platelets interact with the BBB and cerebral parenchyma after injury. It was recently reported that platelet C-type lectin-like 2 (CLEC-2) receptors regulate the neuroinflammatory response and integrity of the BBB following TBI [62]. Platelets can communicate with neurons through release of microparticles, exosomes, and potentially neurogenic factors [63]. Another method of molecular communication is via damage-/pathogen-associated molecular patterns (DAMP/PAMPS) that are released following TBI and induce a platelet-mediated inflammatory response [64]. Platelet accumulation in regions of CNS damage helps promote neurogenesis and white matter repair [65].

It is interesting that decreased OXPHOS rates in platelets at 6 h precedes decreased State III bioenergetics in the cortex at 24 h in the CHI model [2]. Platelets may sense specific neuronal/glial populations near the vasculature to account for this discrepancy in metabolic changes. Previously, it has been reported that maximal respiration of cortical mitochondria from female green monkeys positively correlates with parameters of platelet respiration [29]. Our group has shown dysfunction in brain mitochondria early after both CCI and CHI [2,36]; further analysis will examine the correlation with platelet bioenergetics after experimental TBI. We observe that platelet coupling increases at 24 h after CCI, which coincides with decreased mitochondrial bioenergetics in the brain, as previously described [66]. This is consistent with acutely elevated platelet coupling efficiency corresponding with lower cerebral bioenergetics in a model of cardiac arrest [30].

Platelets are bioenergetically well-coupled compared to other blood cell types, demonstrating higher ATP-linked respiration (OXPHOS) and lower LEAK respiration [42]. Platelets are also more glycolytic compared to other blood cell types [42]. The lack of glycolytic alteration in this current study suggests that platelet bioenergetic changes are occurring at the mitochondrial level. Although no changes were seen in platelet glycolysis early after TBI, it would be important to examine whether the glycolytic response to a cellular challenge (e.g., agonists such as thrombin) is altered after TBI. It is plausible that platelets in TBI subjects could be predisposed to altered metabolic responses. Furthermore, mitochondrial dysfunction in platelets themselves could result in secondary coagulopathy, leading to increased risk of thrombosis through activation and microparticle secretion [67].

Shifts in platelet respiration early after injury could be, in part, due to platelet turnover. However, newly formed platelets may continue to interact with the dysfunctional BBB and demonstrate altered bioenergetics. Alternatively, mitochondrial respiration alterations in platelets could represent changes in the mitochondrial genome [68]. Although there is a lack of investigation regarding mtDNA changes in the context of TBI, platelets could be the ideal sensor to measure these changes.

We acknowledge several limitations of this study. First, only male mice were used in these assays. While in the absence of TBI, platelet respiration in humans does not significantly vary with sex [43], additional studies are needed to assess sex-specific responses in the context of TBI. Female animals should be included in future studies to fully encompass the systemic blood cell response to TBI relative to sex. Blood cell changes were only examined at 6 and 24 h after injury; future studies should examine whether platelet bioenergetics are altered chronically following TBI. While hematological analysis was able to uncover general changes in white blood cell populations after TBI, future studies are needed to perform FACS phenotyping analysis to examine alterations in specific subsets of leukocytes in both whole blood and the spleen after CCI and CHI. There are also limitations to the examination of respiration in intact platelets. Succinate, which has limited cellular uptake, is given to the intact platelets in this study, so it is difficult to know the affect succinate has on platelet respiration [69]. Future studies should examine whether cell permeabilization with direct, unhindered access to mitochondria [43] produces any further insight into TBI-induced alteration of platelet mitochondrial bioenergetics.

In this study, prostacyclin, which inhibits platelet activation and aggregation, was in the buffer during the Seahorse experiments. Therefore, the results are indicative of platelet bioenergetic changes in an inactivated state with potentially altered glucose metabolism. Future studies, incorporating a wash of prostacyclin before respirometry measurements, should be performed. Moreover, platelet function may be altered after TBI with the addition of an activating stimuli, such as ADP or thrombin [70]. Platelets do not have a high reserve respiratory capacity, especially when compared to other blood cells, such as monocytes or lymphocytes [42,43]. Here, we show no significant differences between OXPHOS and MAX levels within our experimental groups. However, Sjovall et al. showed that MAX rates are significantly lower with the addition of oligomycin compared to those without oligomycin [43]. This highlights a potential limitation of the present study, though it is difficult to compare between different respirometry devices.

## 5. Conclusions

Our findings indicate that CCI produces early blood cell alterations, including low blood leukocyte levels, dynamic PNA interactions, and alterations in metabolic coupling in platelets. CHI, however, appears to produce thromboinflammatory effects with high acute levels of blood leukocytes, slightly elevated PNAs, and delayed increases in platelet bioenergetics. These mechanisms, related to thromboinflammation, require further exploration, though current results highlight clinically-relevant outcomes following CHI. Finally, we posit that platelet bioenergetics may have diagnostic and prognostic value, related to cerebral metabolism, to peripherally monitor progression of TBI.

## Figures and Tables

**Figure 1 cells-10-00500-f001:**
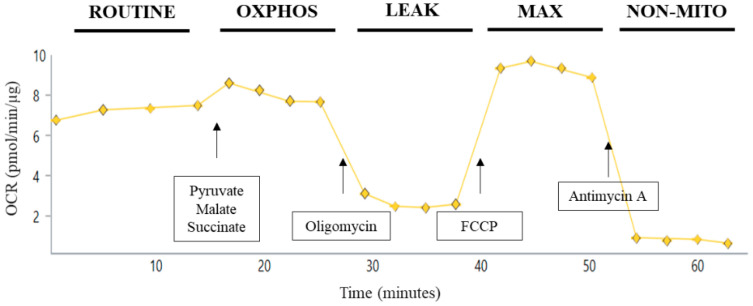
Substrate/inhibitor injection protocol for Seahorse assay of intact platelets. Utilizing the Seahorse XFe96 Flux Analyzer enables the assessment of up to 30 independent samples in triplicate at one time. For respirometry of intact platelets, platelets were added in prostacyclin/phosphate/glucose buffer; this state of respiration is defined as ROUTINE. For injection A, 5 mM pyruvate, 2.5 mM malate, and 10 mM succinate were added to give OXPHOS_CI+CII_ of platelets. Oligomycin (5 μM) was injected next to give LEAK platelet respiration. FCCP (4 µM) was then added in injection C for MAX_CI+CII_. Finally, non-mitochondrial (NON-MITO) oxygen consumption, independent of the electron transport system, is measured after addition of antimycin A (1µm).

**Figure 2 cells-10-00500-f002:**
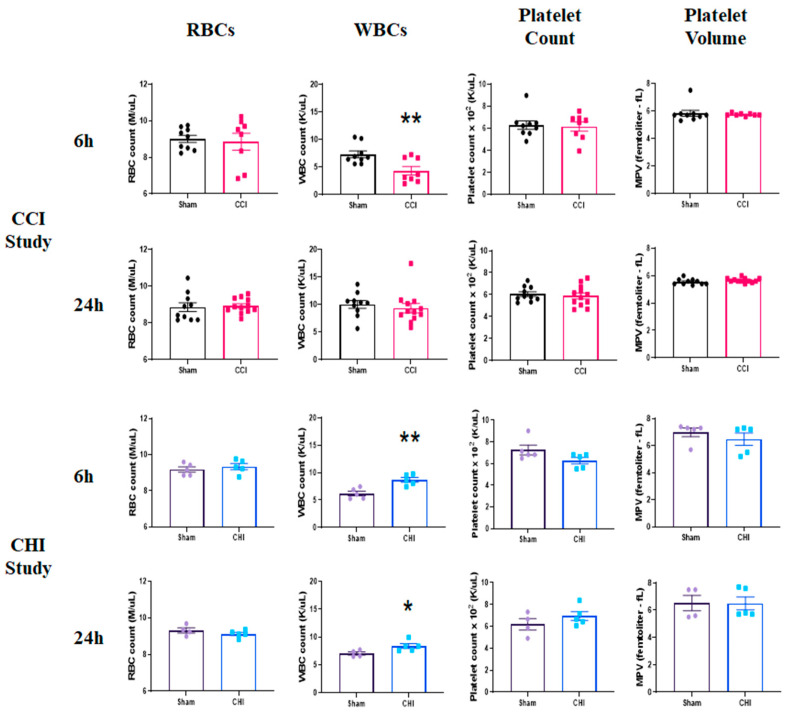
Early profile of blood cell alterations following contusion brain injury (CCI) and mild closed head impact (CHI). (Top Panels) Mice received either sham injury or severe CCI followed by euthanasia at either 6 or 24 h post-injury. Examination of RBC counts, WBC counts, platelet counts, and mean platelet volume (MPV) was performed by an IDEXX ProCyte Dx Analyzer. CCI does not alter RBC counts, platelet counts, or platelet volume at 6 or 24 h after injury. However, WBCs were significantly decreased after CCI compared to sham at 6 h post-injury before restoration at 24 h post-injury. ** *p* = 0.0076; t = 3.083. n = 8–9/group. (Bottom Panels) Mice received either sham injury or mild CHI followed by euthanasia at either 6 or 24 h post-injury. Examination of RBC counts, WBC counts, platelet counts, and mean platelet volume (MPV) was performed. CHI does not significantly change RBC counts, platelet counts, or platelet volume at 6 or 24 h after injury. However, WBCs were significantly increased after CHI compared to sham at both 6 and 24 h post-injury. ** *p* = 0.0039; t = 4.009. * *p* = 0.048; t = 2.397. n = 4–5/group. Mean ± SEM.

**Figure 3 cells-10-00500-f003:**
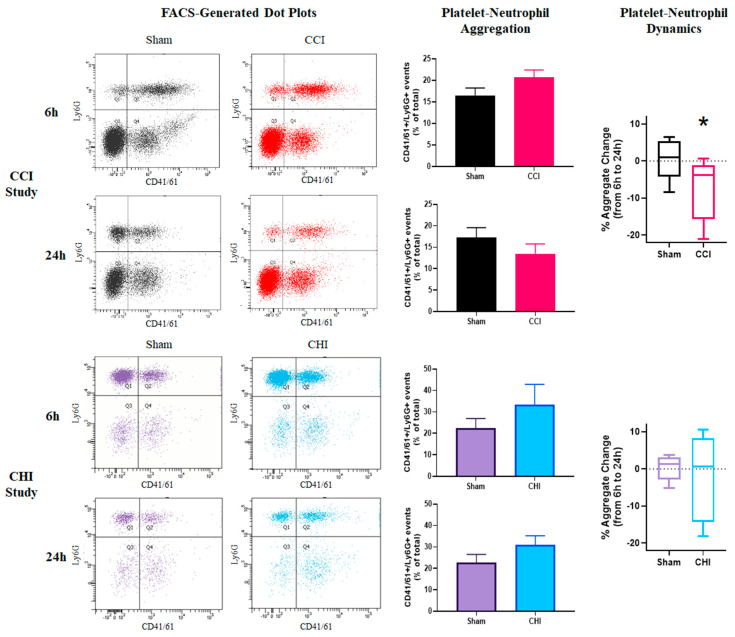
Divergent early profiles of platelet-neutrophil aggregation after contusion brain injury and mild closed head impact. (Top Panels) Mice received either sham injury or severe CCI followed by euthanasia at either 6 or 24 h post-injury. FACS analysis was performed on blood samples to examine platelet-neutrophil aggregation. Representative dot plots of flow cytometry showing platelet-neutrophil aggregates (PNAs) in the top-right quadrant of each plot. There were no significant differences in PNAs after CCI at either 6 or 24 h post-injury when compared to time-matched sham controls. There was a significant difference between sham and CCI groups in platelet-neutrophil dynamics (relative change of PNAs) from 6 to 24 h post-injury. * *p* = 0.032. t = 2.373. n = 8–9/group. (Bottom Panels) Mice received either sham injury or mild CHI followed by euthanasia at either 6 or 24 h post-injury. FACS analysis was performed on blood samples to examine platelet-neutrophil aggregation. Representative dot plots of flow cytometry showing platelet-neutrophil aggregates (PNAs) in the top-right quadrant of each plot. Levels of platelet-neutrophil aggregates (PNAs) was slightly elevated, though not significantly different, in mice receiving CHI at both 6 and 24 h post-injury when compared to time-matched sham controls. There was no significant difference between sham and CHI groups in the platelet-neutrophil dynamics from 6 to 24 h post-injury. n = 5/group. Mean ± SEM.

**Figure 4 cells-10-00500-f004:**
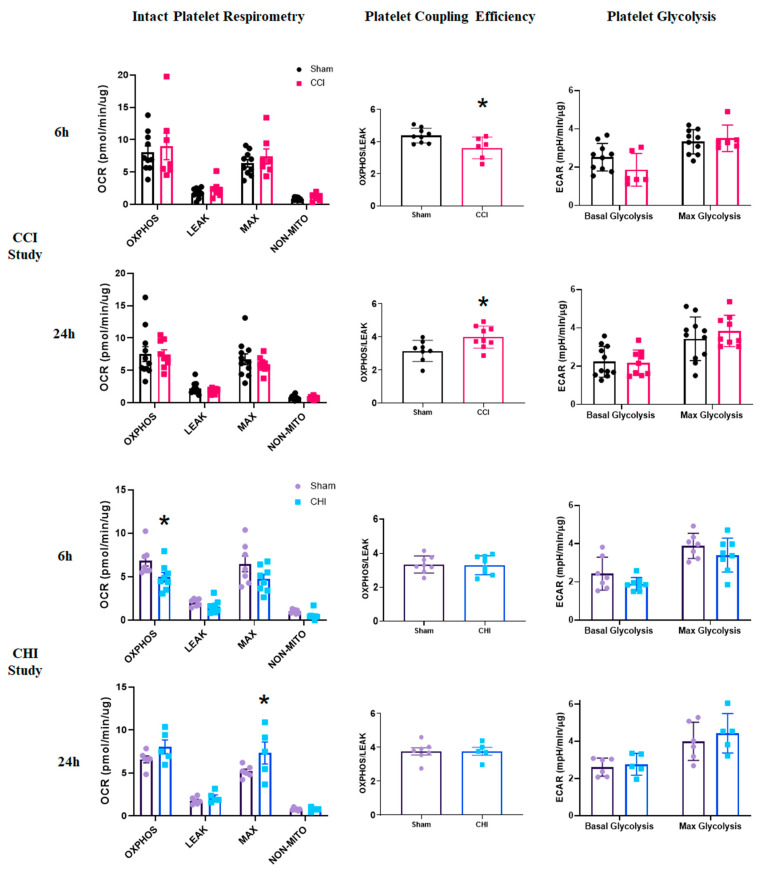
Differential alterations in platelet coupling and oxidative phosphorylation acutely following contusion brain injury and mild closed head injury. (Top Panels) Mice received either sham injury or severe CCI followed by euthanasia at either 6 or 24 h post-injury. Isolated platelets underwent respirometry using the Seahorse XFe96. At 6 or 24 h post-CCI, there were no significant differences in any state of respiration measured. Platelet coupling efficiency was reduced 6 h after CCI compared to sham. t = 2.656; * *p* = 0.012. There was a significant increase in platelet coupling efficiency 24 h after CCI compared to sham. t = 2.637; * *p* = 0.019. Platelet glycolysis was unchanged at 6 and 24 h after CCI. n = 6–10/group, (Bottom Panels) Mice received either sham injury or mild CHI followed by euthanasia at either 6 or 24 h post-injury. Isolated platelets underwent respirometry using the Seahorse XFe96. At 6 h post-CHI, OXPHOS_CI+CII_ was significantly lower in the CHI group compared to sham. t = 2.284; * *p* = 0.040. In general, overall platelet respiration was lower at 6 h after CHI compared to sham. At 24 h, post-CHI, MAX_CI+CII_ was significantly higher in the CHI group compared to sham. t = 3.12; * *p* = 0.0142. In general, overall platelet respiration was higher at 24 h after CHI compared to sham. Platelet coupling efficiency was unchanged at 6 and 24 h after CHI. There was no significant difference in platelet glycolysis at either 6 or 24 h following CHI. n = 5–8/group. Mean ± SEM.

## Data Availability

The data presented in this study are available on request from the corresponding author.

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
