# Peer review of "Differential Leukocyte and Platelet Profiles in Distinct Models of Traumatic Brain Injury"

_cells, 2021, doi:10.3390/cells10030500_

Round 1
Reviewer 1 Report
Hubbard et al investigated the dynamics and metabolic status of platelets and temporally profiled the peripheral leukocytes number after experimental mild traumatic brain injury (mTBI). The authors also compared the difference in the above pathological indexes between the controlled cortical impact (CCI) and closed head injury (CHI) preclinical TBI models. They found that CCI induced early reduction of leukocyte counts in blood at 6h whereas an CHI increased the leukocyte count at 6 and 24 hours after an CHI. While both the total platelet counts and volume did not change after CCI and CHI, a decreased platelet-neutrophil aggregation (PNAs), an indicator of thromboinflammatory, was observed in the CCI mice. They further showed that the level of platelet mitochondrial oxidative phosphorylation (OXPHOS) was decreased at 6h and increased later at 24h after CCI. They concluded that platelet bioenergetics could be used as a risk and prognostic prediction tool in a clinical setting TBI. The characteristic and metabolic changes in platelets after TBI have not been previously reported, making this study novel. However, there are major flaws in the research design that largely reduce the reviewer’s enthusiasm of the current work. The comments are offered as follows: 1. The authors reported the number of blood cells based on results from the hematology analyzer, which is inadequate in the reviewer’s opinion. The authors should use flow cytometry analysis instead to determine the composition and number of blood cells in the whole blood and spleen samples after CCI and CHI. 2. Since the aim of the current study was to compare the divergent immune cell profile and platelet bioenergetics between CCI and CHI, the authors should integrate the results from two models in one plot or combine them into a single figure for ease of understanding. 3. The results showing decreased platelet-neutrophil aggregation from flow cytometry analysis are unconvincing. Since cell debris/dead cells can also aggregate and bind to the antibodies, the authors should gate on live cells and select for the Ly6G and CD41/61 double positive population. The gating strategy should be provided as supplementary. 4. The authors claimed that the status of platelet bioenergetics might have translational potential as a diagnostic and prognostic tool for TBI. However, they did not provide evidence showing the correlation between the level of platelet OXPHOS, contusion volume, and long-term neurobehavioral outcomes of TBI. They should also provide background regarding the correlation between platelet metabolic status and their functions to help unfamiliar audience understand the underlying rationale of their experimental design.Author Response
We thank the reviewers for their helpful and thoughtful comments and have addressed each question or comment in the revised manuscript. Our responses are also briefly summarized (in italics) below.
Hubbard et al investigated the dynamics and metabolic status of platelets and temporally profiled the peripheral leukocytes number after experimental mild traumatic brain injury (mTBI). The authors also compared the difference in the above pathological indexes between the controlled cortical impact (CCI) and closed head injury (CHI) preclinical TBI models. They found that CCI induced early reduction of leukocyte counts in blood at 6h whereas a CHI increased the leukocyte count at 6 and 24 hours after an CHI. While both the total platelet counts and volume did not change after CCI and CHI, a decreased platelet-neutrophil aggregation (PNAs), an indicator of thromboinflammatory, was observed in the CCI mice. They further showed that the level of platelet mitochondrial oxidative phosphorylation (OXPHOS) was decreased at 6h and increased later at 24h after CCI. They concluded that platelet bioenergetics could be used as a risk and prognostic prediction tool in a clinical setting TBI. The characteristic and metabolic changes in platelets after TBI have not been previously reported, making this study novel. However, there are major flaws in the research design that largely reduce the reviewer’s enthusiasm of the current work. The comments are offered as follows:
The authors thank the reviewer for the feedback and hope the revisions improve the manuscript.
- The authors reported the number of blood cells based on results from the hematology analyzer, which is inadequate in the reviewer’s opinion. The authors should use flow cytometry analysis instead to determine the composition and number of blood cells in the whole blood and spleen samples after CCI and CHI.
We absolutely agree with the reviewer’s point and will pursue a more detailed analysis of different leukocyte populations in our future studies. In this manuscript, we focused on platelet activation, and measured platelet-neutrophil aggregates, which is a clear indication of platelet activation since platelet exposure of P-Selectin is critical for this interaction. With our focus on platelets as a potential diagnostic biomarker to distinguish between CCI and CHI, hematology parameters were used to examine overt platelet changes and the Hematology Analyzer is well suited to evaluate platelet counts due to its ease, accuracy, and consistency. The changes in WBCs uncovered by our analysis are exciting and will be pursued in the future using FACS analysis with phenotyping panels to define which leukocyte subtypes are altered during mTBI (line 479-482 in Discussion). Future studies, beyond the scope of the current manuscript, will seek to determine composition of spleen samples to further shed light on hematological changes and outcomes post CCI and CHI.
- Since the aim of the current study was to compare the divergent immune cell profile and platelet bioenergetics between CCI and CHI, the authors should integrate the results from two models in one plot or combine them into a single figure for ease of understanding.
Thank you for this comment and we do agree that comparison between CCI and CHI would be made easier if these were on the same figures. Therefore, we have re-structured the figures to include outcome measures from both TBI models.
- The results showing decreased platelet-neutrophil aggregation from flow cytometry analysis are unconvincing. Since cell debris/dead cells can also aggregate and bind to the antibodies, the authors should gate on live cells and select for the Ly6G and CD41/61 double positive population. The gating strategy should be provided as supplementary.
The flow cytometry analysis was indeed performed on live cells and this has been made clear in the Methods (line 205-206). The reviewer is correct regarding cell debris/dead cells aggregating and thus, we have selected for only Ly6G and CD41/61 double-positive events. We have provided the gating strategy as Fig. A2.
- The authors claimed that the status of platelet bioenergetics might have translational potential as a diagnostic and prognostic tool for TBI. However, they did not provide evidence showing the correlation between the level of platelet OXPHOS, contusion volume, and long-term neurobehavioral outcomes of TBI. They should also provide background regarding the correlation between platelet metabolic status and their functions to help unfamiliar audience understand the underlying rationale of their experimental design.
We appreciate the suggestion. Since mice have a limited blood volume blood, draws for biochemical assays are terminal procedures and thus we cannot perform “within animal” correlation analyses. Though we feel this is a next step for further translation, it is beyond the scope of our current technologies. We are working on FACS-based technologies that may allow us to evaluate mitochondrial function in the platelets from smaller volumes of blood (tail nick or retro-orbital bleed). Once we can perform these studies, mice will be frequently monitored for platelet status post-injury followed by neuro-behavioral and histological analysis in future investigation (line 390-391). We discuss how platelet metabolism relates to platelet activation, which in turns plays a vastly important role in mechanisms of systemic thrombosis (line 427-434). Further, platelet crosstalk with brain cells could identify platelet metabolism as a peripheral monitor for brain metabolism, especially after brain injury (line 435-448).
Reviewer 2 Report
Hubbard and colleagues evaluate leukocyte and platelet profiles at 6 and 24 hours in two rat models of TBI: cortical impact (CCI) and closed head injury (CHD). Widely applicable blood biomarkers to clinically assess severity and type of TBI do not yet exist. Therefore, the differential effects on leukocyte and platelet responses observed after the two TBI models serve as a start to move forward in this arena. Of note in the data are that signatures appear that differentiate the two models with respect mostly to platelet responses. The CHD model is most closely relevant to most “real-world” TBI scenarios and it is valuable that this study provides a beginning for the development of biomarkers of this condition using blood measures. The methods used appear to be solid and comprehensive, encompassing approaches that implicate mitochondrial dysfunctions in blood elements that are unique to the form of TBI.
There are deficiencies over and above those noted by the authors in the discussion. There is an attempt to relate literature findings for each form of TBI to prior observations in blood and those presented here. Of course, it would have been more desirable to have actually examined the brains from each type of TBI to see if unique changes might have pointed to changes in leukocytes and platelets. Such studies may have been beyond the scope of those undertaken in the current work but the authors would be advised to conduct a neurohistological examination in the future to include comprehensive evaluations of microglia, astroglia and myelin markers along with neural (neuronal and glial) injury responses to each type of TBI. Not to be overlooked in these studies would be an examination of the neurovascular unit.
The neuroinflammatory responses in blood alluded to in the present investigation long have been associated with the neurovascular unit in TBI models and in humans. You will be able to find many references on this topic and a paragraph should be devoted to this topic in the discussion.
Finally, some discussion may be warranted regarding the potential for RNA-seq as an approach for biomarker discovery in the blood transcriptome. Several treatments damaging the brain have shown the utility of this approach for analyzing signatures in blood that reflect actions in brain (e.g. see Bowyer et al., PLOS One 2015 Jul 15;10(7):e0133315. doi: 10.1371/journal.pone.0133315).
Author Response
We thank the reviewers for their helpful and thoughtful comments and have addressed each question or comment in the revised manuscript. Our responses are also briefly summarized (in italics) below.
Hubbard and colleagues evaluate leukocyte and platelet profiles at 6 and 24 hours in two rat models of TBI: cortical impact (CCI) and closed head injury (CHD). Widely applicable blood biomarkers to clinically assess severity and type of TBI do not yet exist. Therefore, the differential effects on leukocyte and platelet responses observed after the two TBI models serve as a start to move forward in this arena. Of note in the data are that signatures appear that differentiate the two models with respect mostly to platelet responses. The CHD model is most closely relevant to most “real-world” TBI scenarios and it is valuable that this study provides a beginning for the development of biomarkers of this condition using blood measures. The methods used appear to be solid and comprehensive, encompassing approaches that implicate mitochondrial dysfunctions in blood elements that are unique to the form of TBI.
We thank the reviewer for the helpful feedback.
There are deficiencies over and above those noted by the authors in the discussion. There is an attempt to relate literature findings for each form of TBI to prior observations in blood and those presented here. Of course, it would have been more desirable to have actually examined the brains from each type of TBI to see if unique changes might have pointed to changes in leukocytes and platelets. Such studies may have been beyond the scope of those undertaken in the current work but the authors would be advised to conduct a neurohistological examination in the future to include comprehensive evaluations of microglia, astroglia and myelin markers along with neural (neuronal and glial) injury responses to each type of TBI. Not to be overlooked in these studies would be an examination of the neurovascular unit.
We appreciate the comments, and we agree that it would be helpful to examine correlative measurements of changes in blood cells that what pathological changes are occurring in the brain. By far, the most informative would be to examine mitochondrial bioenergetics in the injured cortex as that would closely relate to platelet bioenergetics. However, when we performed this experiment, we found that blood extraction before decapitation effectively caused significant mitochondrial damage that would not be physiologically relevant. Our group has previously published neurohistological evaluation of both CHI and CCI models (Bolton, 2014; Hubbard, 2019; Pleasant, 2011). Indeed, major pathology of these models has been defined by TBI literature over the past decades. We now introduce the models in better detail, how they relate to human TBI and cite the relevant studies (line 97-111).
The neuroinflammatory responses in blood alluded to in the present investigation long have been associated with the neurovascular unit in TBI models and in humans. You will be able to find many references on this topic and a paragraph should be devoted to this topic in the discussion.
We have now added this in the Discussion (line 392-396).
Finally, some discussion may be warranted regarding the potential for RNA-seq as an approach for biomarker discovery in the blood transcriptome. Several treatments damaging the brain have shown the utility of this approach for analyzing signatures in blood that reflect actions in brain (e.g. see Bowyer et al., PLOS One 2015 Jul 15;10(7):e0133315. doi: 10.1371/journal.pone.0133315).
We have added a brief sentence regarding this (line 419-420). One limitation of doing RNA-seq is obtaining adequate platelet counts from mice. RNA-Seq is viable for human studies as adequate blood volume can be taken from patients but is more technically problematic in mouse samples due to low blood volume.
Reviewer 3 Report
In this manuscript, the authors W. Brad Hubbard, et al., using 2 mouse model for TBI , identify differential leukocyte and platelet profiles.
These findings are novel, exciting, but the authors should perform some additional experiments.
- The authors showed the temporal change in platelet-neutrophil aggregation after TBI only by flow cytometry. These data should be supported by immunofluorescence staining with specific marker.
- In the present study the authors report that “Platelet oxidative phosphorylation is altered acutely after mild closed head injury”. However is highly known that the macrophages can interact with platelets to regulate their redox status by inflammasome activation. Indeed macrophage depletion can diminished platelet-neutrophil aggregation.
Can be possible that the dynamic alterations in platelet bioenergetics depend by macrophage?
The authors should consider this hypothesis.
Author Response
We thank the reviewers for their helpful and thoughtful comments and have addressed each question or comment in the revised manuscript. Our responses are also briefly summarized (in italics) below.
In this manuscript, the authors W. Brad Hubbard, et al., using 2 mouse model for TBI , identify differential leukocyte and platelet profiles.
These findings are novel, exciting, but the authors should perform some additional experiments.
- The authors showed the temporal change in platelet-neutrophil aggregation after TBI only by flow cytometry. These data should be supported by immunofluorescence staining with specific marker.
To our knowledge, immunofluorescence staining and subsequent imaging is not typically used for this type of blood analysis. FACS analysis, which we use, is the standard in the field. To increase transparency concerning the FACS analysis, we now show the gating strategy (Fig. A2).
- In the present study the authors report that “Platelet oxidative phosphorylation is altered acutely after mild closed head injury”. However is highly known that the macrophages can interact with platelets to regulate their redox status by inflammasome activation. Indeed macrophage depletion can diminished platelet-neutrophil aggregation. Can be possible that the dynamic alterations in platelet bioenergetics depend by macrophage? The authors should consider this hypothesis.
Macrophage activation after TBI typically is later than that of neutrophils (Needham, 2019; Schwarzmaier, 2013), so we would expect macrophages to have a lesser role at 6h post-injury. While macrophage activation was not a focus of this manuscript, we now add a sentence highlighting the possibility of macrophage-platelet interaction (line 405-407).
Reviewer 4 Report
This is a well written, well executed study from Hubbard et al. Studying platelet bioenergetics in the context of TBI is a novel and potentially impactful endeavor, and the technical accomplishments and findings from this study will empower future studies. They showed an appreciation for the value of scientific rigor by applying a priori power analyses, randomized enrollment, and blinded analyses, and although they did not address sex as a variable in this initial study, they acknowledged as much and spoke to the importance of doing so in future studies. While I do not see the need for any additional experiments for publication, there are some textual revisions needed throughout the manuscript, and some underlying misinterpretations of CCI-induced leukopenia need to be resolved.
Line 88: There seems to be a logical gap here. It currently reads as if you are positing that because various pathologies can lead to the same GCS score at the system level, we need more system level markers, which obviously isn’t the point that you’re trying to drive home. It could help to explicitly state that TBI pathology and secondary injury consist of multiple endophenotypes (including bioenergetics and thromboinflammation), and that more specific biomarkers are needed to monitor those endophenotypes. You can also revise line 88 from "systematically track neurologic injury" to "systematically track specific aspects of neurologic injury".
Line 96: "further [OUR] understanding"
Line 154: Were there any significant differences in apnea or time to right between sham and injured groups? Was this measured after CCI as well? Negative data are informative, and a lack of effect should be reported. This doesn’t require its own figure but if you’re saying you measured it you should include it in the results.
Lines 204-206: This appears to be significant, and worth restating in the discussion or the associated results section.
Lines 266-268: While adhesion and infiltration are supported by these references (9, 40), they did not observe decreased leukocyte counts. Clinical reports suggest that leukocyte counts increase after TBI, and that this increase correlates with severity of injury (ref 45; Rovlias 2001). Your data suggest that CCI may be suboptimal for studying thromboinflammation after TBI, since leukocyte effect is opposite what is observed clinically. This point should be made here and/or in the Discussion, and could be phrased to demonstrate the benefit of utilizing the CHI model. The suggestion that the decreased leukocyte count is "in corroboration" to reports of increased adherence and brain infiltration is a misinterpretation and should be removed.
Lines 368-369: While Kramer et al. (43) showed that, unrelated to the current statement, experimental depletion of Tregs can result in increased leukocyte infiltration in the brain, this in no way suggests that leukopenia can be attributed to increased brain infiltration of leukocytes. None of the other papers cited establish this connection either. The suggestion that leukopenia could be due to increased brain infiltration is not supported by the literature, and directly contradicts clinical observations of both leukocytosis and increased brain infiltration together in human TBI. The leukopenia observed after CCI is difficult to explain, but it is not due to increased infiltration, and it clearly indicates a disconnect from the mechanisms and manifestations of human TBI. Possibly the most important finding in this report is that CHI reproduces leukocytosis observed in clinical TBI while CCI does not. This needs to be emphasized and explored in the discussion, while the spurious suggestion that leukopenia can be due to increased brain infiltration must be scrubbed from this manuscript.
Lines 371-372: As written this seems to imply that WBCs were measured at 6 months. Please revise to indicate that blood leukocyte levels were correlated with GCS scores shortly after hospital admission and further demonstrated prognostic potential as they were also associated with recovery outcome 6 months later.
Line 386: In addition to extended time course, multiple CHI severities could provide insight into diagnostic/prognostic potential.
The conclusion should include your notable finding that leukocytosis in response to CHI was similar to what is observed after human TBI, while the CCI leukopenia response diverged significantly from human TBI.
Finally, CCI and CHI may offer the ability to study differences in the effects of focal vs diffuse brain injuries, but it should be clearly stated that both still induce injury through compressive mechanical force, whereas diffuse injury in humans and other large-brained animals is induced primarily through diffuse shear deformation forces generated by rapid acceleration/deceleration of the head. There are large differences in the consequences of compression-induced damage versus the clinical manifestations of shear deformation-induced damage, including but not limited to neurological outcomes (e.g., loss of consciousness), gross pathology (e.g., bridging vein rupture and regional patterns of damage to vasculature and white matter tracts), and substrates/distribution of cell and molecular pathology. Given the importance placed upon testing the consequences of different injury inductions in this study, the difference between compression and shear deformation injury should be clearly stated to avoid contributing to misinterpretation of the results by readers. While this in no way detracts from studies utilizing compressive injuries, it is vitally important to avoid overstating the scope of TBI mechanisms covered by these two compression injury models or their similarity to human TBI, which could inadvertently lead readers to overestimate the translational potential for compression injury models. Models and studies do not need to be IND-enabling to be highly relevant, and the translational pipeline for neurotrauma is a bit of a mess, so revisions that clarify these points are important to provide clarity that will help improve the dismal translational record in neurotrauma.
Author Response
We thank the reviewers for their helpful and thoughtful comments and have addressed each question or comment in the revised manuscript. Our responses are also briefly summarized (in italics) below.
This is a well written, well executed study from Hubbard et al. Studying platelet bioenergetics in the context of TBI is a novel and potentially impactful endeavor, and the technical accomplishments and findings from this study will empower future studies. They showed an appreciation for the value of scientific rigor by applying a priori power analyses, randomized enrollment, and blinded analyses, and although they did not address sex as a variable in this initial study, they acknowledged as much and spoke to the importance of doing so in future studies. While I do not see the need for any additional experiments for publication, there are some textual revisions needed throughout the manuscript, and some underlying misinterpretations of CCI-induced leukopenia need to be resolved.
We genuinely appreciate the kind comments and suggestions to improve the manuscript.
Line 88: There seems to be a logical gap here. It currently reads as if you are positing that because various pathologies can lead to the same GCS score at the system level, we need more system level markers, which obviously isn’t the point that you’re trying to drive home. It could help to explicitly state that TBI pathology and secondary injury consist of multiple endophenotypes (including bioenergetics and thromboinflammation), and that more specific biomarkers are needed to monitor those endophenotypes. You can also revise line 88 from "systematically track neurologic injury" to "systematically track specific aspects of neurologic injury".
Thank you for the distinction and this has been edited (line 84-87).
Line 96: "further [OUR] understanding"
This has been changed (line 95).
Line 154: Were there any significant differences in apnea or time to right between sham and injured groups? Was this measured after CCI as well? Negative data are informative, and a lack of effect should be reported. This doesn’t require its own figure but if you’re saying you measured it you should include it in the results.
Yes, there were significant differences in both apnea and time to right between injured and sham groups in the CHI model. This has been added to the Results (line 284-286). Righting reflex was not measured following CCI due to craniotomy procedure; this is consistent with the vast majority of CCI studies [1]. From our experience, lateral CCI does not cause an apnea response.
Lines 204-206: This appears to be significant, and worth restating in the discussion or the associated results section.
We agree that our novel procedure in assessing platelet respiration from a single mouse is a significant leap forward of this study. We have re-stated this is the Results (line 338-340).
Lines 266-268: While adhesion and infiltration are supported by these references (9, 40), they did not observe decreased leukocyte counts. Clinical reports suggest that leukocyte counts increase after TBI, and that this increase correlates with severity of injury (ref 45; Rovlias 2001). Your data suggest that CCI may be suboptimal for studying thromboinflammation after TBI, since leukocyte effect is opposite what is observed clinically. This point should be made here and/or in the Discussion, and could be phrased to demonstrate the benefit of utilizing the CHI model. The suggestion that the decreased leukocyte count is "in corroboration" to reports of increased adherence and brain infiltration is a misinterpretation and should be removed.
While the limited breadth of the study at hand argues that CHI may be a better representation of human TBI in terms of leukocyte counts, we feel that, in general, it is an oversimplification to suggest this in the Discussion. Considering the temporal aspects of early mechanisms after brain injury and the difficulty translating “mouse time” to “human time,” there is still much to be understood regarding thromboinflammation. There is a lack of studies in the field and we hope this encourages more investigation by other researchers. We have provided more comparison to these clinical studies (line 380-385). We do agree with that blood leukocyte decrease does not corroborate reports showing increased leukocyte infiltration in the brain and have removed this discussion point.
Lines 368-369: While Kramer et al. (43) showed that, unrelated to the current statement, experimental depletion of Tregs can result in increased leukocyte infiltration in the brain, this in no way suggests that leukopenia can be attributed to increased brain infiltration of leukocytes. None of the other papers cited establish this connection either. The suggestion that leukopenia could be due to increased brain infiltration is not supported by the literature, and directly contradicts clinical observations of both leukocytosis and increased brain infiltration together in human TBI. The leukopenia observed after CCI is difficult to explain, but it is not due to increased infiltration, and it clearly indicates a disconnect from the mechanisms and manifestations of human TBI. Possibly the most important finding in this report is that CHI reproduces leukocytosis observed in clinical TBI while CCI does not. This needs to be emphasized and explored in the discussion, while the spurious suggestion that leukopenia can be due to increased brain infiltration must be scrubbed from this manuscript.
Thank you for pointing this out; we previously did not appreciate that there was consensus that leukocytosis occurs after human TBI. Considering the clinical observations of both leukocytosis and increased brain infiltration, we agree that the argument is a non-starter. As stated in response to the previous comment, we have removed all statements related to the decrease in blood leukocytes levels after CCI attributed to increased brain infiltration and discuss the clinical relevance of increased leukocyte levels after CHI (line 390).
Lines 371-372: As written this seems to imply that WBCs were measured at 6 months. Please revise to indicate that blood leukocyte levels were correlated with GCS scores shortly after hospital admission and further demonstrated prognostic potential as they were also associated with recovery outcome 6 months later.
This has been revised.
Line 386: In addition to extended time course, multiple CHI severities could provide insight into diagnostic/prognostic potential.
The conclusion should include your notable finding that leukocytosis in response to CHI was similar to what is observed after human TBI, while the CCI leukopenia response diverged significantly from human TBI.
While producing varying severities in CHI models is not commonly seen, repeated CHIs could be used to produce exacerbated/prolonged outcomes and examine diagnostic/prognostic potential. Again, we comment on the relevance of CHI without overstating based on limited results. CHI does not produce any sub/epidural hemorrhage or contusion so it has limited applicability to human TBI involves hemorrhage as relates to thrombosis. We expand on pathology of CCI and CHI as brought up also by Reviewer 2.
Finally, CCI and CHI may offer the ability to study differences in the effects of focal vs diffuse brain injuries, but it should be clearly stated that both still induce injury through compressive mechanical force, whereas diffuse injury in humans and other large-brained animals is induced primarily through diffuse shear deformation forces generated by rapid acceleration/deceleration of the head. There are large differences in the consequences of compression-induced damage versus the clinical manifestations of shear deformation-induced damage, including but not limited to neurological outcomes (e.g., loss of consciousness), gross pathology (e.g., bridging vein rupture and regional patterns of damage to vasculature and white matter tracts), and substrates/distribution of cell and molecular pathology. Given the importance placed upon testing the consequences of different injury inductions in this study, the difference between compression and shear deformation injury should be clearly stated to avoid contributing to misinterpretation of the results by readers. While this in no way detracts from studies utilizing compressive injuries, it is vitally important to avoid overstating the scope of TBI mechanisms covered by these two compression injury models or their similarity to human TBI, which could inadvertently lead readers to overestimate the translational potential for compression injury models. Models and studies do not need to be IND-enabling to be highly relevant, and the translational pipeline for neurotrauma is a bit of a mess, so revisions that clarify these points are important to provide clarity that will help improve the dismal translational record in neurotrauma.
Thank for this elegant explanation and we now highlight the differences in brain injury mechanics. We wholly agree that caution should be exercised not to overstate or overinterpret findings and hope the revisions put this in perspective.
- Siebold, L., A. Obenaus, and R. Goyal, Criteria to define mild, moderate, and severe traumatic brain injury in the mouse controlled cortical impact model. Experimental Neurology, 2018. 310: p. 48-57.
Round 2
Reviewer 1 Report
The authors have adequately addressed reviewer's concerns and the current manuscript has been largely improved and is suitable for publication.
Reviewer 3 Report
The authors answered all my critical points. I propose to publish this interesting paper in cells journal.